# AHR:IKAROS Interaction Promotes Platelet Biogenesis in Response to SR1

Lea Mallo [1],[†], Valentin Do Sacramento [1],[†], Christian Gachet [1], Susan Chan [2], Philippe Kastner [2], François Lanza [1], Henri de la Salle [1] and Catherine Strassel [1],*

1 Institut National de la Santé et de la Recherche Médicale (INSERM), Etablissement Français du Sang (EFS)-Grand Est, Biologie et Pharmacologie des Plaquettes Sanguines (BPPS) Unité Mixte de Recherche (UMR)_S1225, Fédération de Médecine Translationnelle de Strasbourg (FMTS), Université de Strasbourg, F-67000 Strasbourg, France; lea.mallo@efs.sante.fr (L.M.); valentin.dosacramento@efs.sante.fr (V.D.S.); christian.gachet@efs.sante.fr (C.G.); francois.lanza@efs.sante.fr (F.L.); henri.delasalle@efs.sante.fr (H.d.l.S.)
2 Institut de Génétique et de Biologie Moléculaire et Cellulaire (IGBMC), INSERM U1258, Centre National de la Recherche Scientifique (CNRS), Unité Mixte de Recherche (UMR) 7104, Université de Strasbourg, F-67404 Illkirch, France; chan@igbmc.fr (S.C.); kastner@igbmc.fr (P.K.)
* Correspondence: catherine.strassel@efs.sante.fr; Tel.: +33-388-21-25-25; Fax: +33-388-21-25-21
† Both authors contributed equally to this study and were considered as co-first authors.

**Abstract:** In vitro, the differentiation of megakaryocytes (MKs) is improved by aryl-hydrocarbon receptor (AHR) antagonists such as StemRegenin 1 (SR1), an effect physiologically recapitulated by the presence of stromal mesenchymal cells (MSC). This inhibition promotes the amplification of a CD34$^+$CD41$^{low}$ population able to mature as MKs with a high capacity for platelet production. In this short report, we showed that the emergence of the thrombocytogenic precursors and the enhancement of platelet production triggered by SR1 involved IKAROS. The downregulation/inhibition of IKAROS (shRNA or lenalidomide) significantly reduced the emergence of SR1-induced thrombocytogenic population, suggesting a crosstalk between AHR and IKAROS. Interestingly, using a proximity ligation assay, we could demonstrate a physical interaction between AHR and IKAROS. This interaction was also observed in the megakaryocytic cells differentiated in the presence of MSCs. In conclusion, our study revealed a previously unknown AHR/ IKAROS -dependent pathway which prompted the expansion of the thrombocytogenic precursors. This AHR- IKAROS dependent checkpoint controlling MK maturation opens new perspectives to platelet production engineering.

**Keywords:** AHR; IKAROS; megakaryocytes; platelet production





## 1. Introduction

Blood platelets are efficiently generated from bone marrow megakaryocytes (MKs) where each MK is predicted to produce 2000–3000 platelets. Despite recent progress, platelet production is only reproduced with low efficiency under in vitro culture conditions. In view of the increasing demand for platelet concentrates that are free of infectious, inflammatory and immune risks, the production of cultured platelets still needs to be greatly improved to become a promising transfusion product [1]. It is therefore of utmost importance to fully clarify the molecular and cellular mechanisms underlying their biogenesis.

Megakaryopoiesis (MKP) is a hierarchized and complex process where hematopoietic stem cells (HSCs) differentiate to give rise to megakaryocytic progenitors, which will differentiate into mature MKs capable of extending cytoplasmic extensions into the sinusoid vessel to release platelets under flow [2]. These successive cell fate decisions are under the coordination of an array of extrinsic differentiation or growth factor, small metabolites, stromal mesenchymal cells (MSCs), and intrinsic transcription factors (TFs) [3–5]. In a previous study, we reported that the Aryl-Hydrocarbon Receptor (AHR) antagonist, StemRegenin 1 (SR1), favors the expansion in culture at day 10 of a specific population of

progenitors, phenotypically defined as CD34$^+$CD41$^{low}$ cells, empowered with an enhanced potential to generate mature MKs allowing us to obtain a yield of 50–100 platelets/CD34 progenitor [6]. At this point the AHR-dependent mechanisms involved in specifying this remarkable population remain unresolved.

## 2. Results and Discussion

The observation that CD34$^+$CD41$^{low}$ cells harbor signs of an immature state, illustrated by low demarcation membrane system amplification and ploidy, raised the hypothesis of MK differentiation inhibition at a transcriptional level through SR1-modulated AHR. A number of hematopoietic TFs have been identified to promote or favor MK progenitor specification, MK maturation and platelet formation, including GATA-1, Friend of GATA-1 (FOG-1, ZFPM1), Fli-1, RUNX-1 and NF-E2 [7]. These TFs have also been described to promote the expression of megakaryocytic receptors [3–5]. In contrast, IKAROS (IKZF1), a member of the zing-finger protein family important for early lymphocyte development, has been shown to exert a repressor effect on MK differentiation and to restrain terminal MK maturation [8,9]. Indeed, some effector pathways downstream from IKAROS are involved in MKP such as CTNND1 (catenin delta 1/p120 catenin), belonging to the Wnt, Rho family GTP-ase, CDKN1A (cyclin dependent kinase inhibitor 1A). These target genes are crucial actors of MK maturation and platelet production. This effect is thought to occur via its interaction with a network of TFs and TF-associated proteins including GATA and NF-E2. Interestingly, it was recently observed that IKAROS negatively regulates the immune response of the gut by interacting with AHR. These findings therefore suggest the hypothesis that IKZF1 could represent a molecular link in CD34$^+$CD41$^{low}$ cells expansion in response to the AHR antagonist SR1 [10].

To test this hypothesis, we first refined the phenotype of CD34$^+$CD41$^{low}$ cells by evaluating whether the expression of tetraspanin CD9, a marker of mature MKs [11], could represent another differentiating criteria. Flow cytometry analysis of CD9 expression within the CD34$^+$CD41$^{low}$ population allowed delineating two subpopulations: CD34$^+$CD41$^+$CD9$^+$ and CD34$^+$CD41$^+$D9$^-$ cells (Figure 1Ai), representing 41.4 ± 0.6% and 58.3 ± 0.7% of the CD34$^+$CD41$^{low}$ cells, respectively (Figure 1Aii). We then evaluated the capacities of these two subpopulations to produce mature MKs and platelets following cell sorting and culture in the presence of TPO and SR1 (Figure 1Bi), as described previously by our group [6]. Whereas less than 10% of CD34$^+$CD41$^{low}$CD9$^+$-derived MKs exhibited proplatelets, this proportion rose to more than 90% when MKs were derived from CD34$^+$CD41$^{low}$CD9$^-$ cells (Figure 1Bii). This was accompanied by a dramatic difference in platelet production between the two subpopulations (0.6 ± 0.1 × 10$^6$/well vs. 1.85 ± 0.3 × 10$^6$/well, respectively) (Figure 1Biii). This represented a slightly higher efficiency than with the whole CD34$^+$CD41$^{low}$ population (1.39 ± 0.24 × 10$^6$ /well), suggesting a negative effect of the CD9$^+$ subpopulation on the potential of the CD9$^-$ subpopulation to support platelet production. Using this CD34$^+$CD41$^+$CD9$^-$ phenotype to define the population of interest (POI) we then investigated the role of IKAROS in its expansion. We compared the POI to CD34$^-$CD41$^+$ cells, previously shown to represent more mature MKs, based on their ploidy and DMS development (mature MK, MMK) but unable to produce proplatelets when passed in TPO-containing media [6]. qRT-PCR revealed a reduced level of expression of a number of genes representative of MK maturation (CD9, PF4, GP1BA, GATA1, NFE2) supporting the hypothesis of a negative role of IKFZ1 on the MK maturation in response to SR1 [9] (Figure S1).

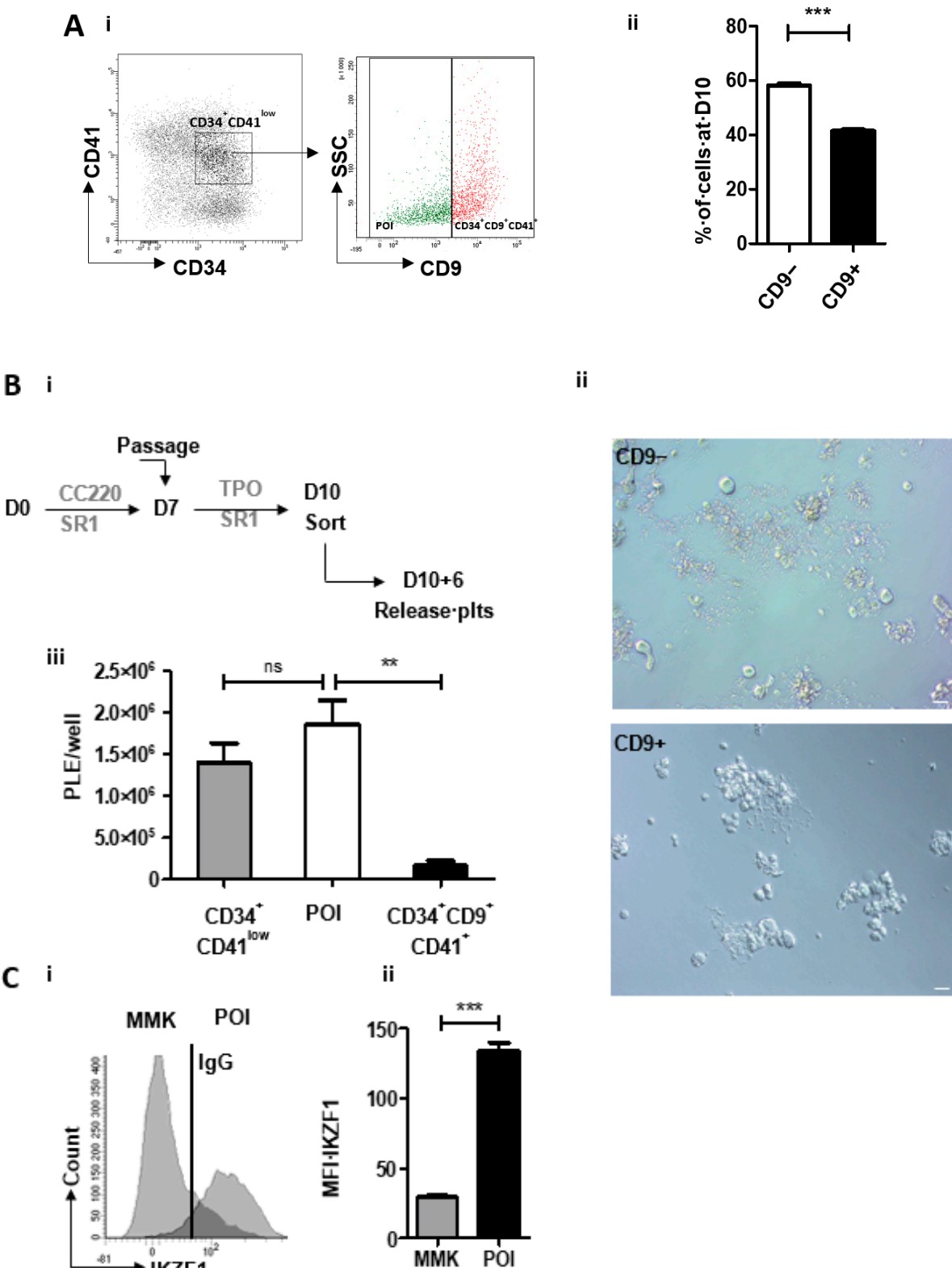

**Figure 1.** SR1-derived population of interest (POI) expresses IKAROS. (**A**) *Refining the CD34⁺CD41$^{low}$ cells phenotype as CD34⁺CD41⁺CD9⁻*. i. Expression of CD34 and CD41 markers at day 10 in SR1 culture conditions (left). Flow cytometric analysis of side scatter and CD9 in CD34⁺CD41$^{low}$ cells identifying two subpopulations based on CD9 expression (right). ii. analysis of the percentage of CD9⁺ and CD9⁻ subpopulations in CD34⁺CD41$^{low}$ cells (41.4 ± 0.6 CD9⁺ vs. 58.3±0.7 CD9⁻). (mean ± SEM, n = 5, *** $p < 0.001$, Student $t$ test). (**B**) *Analysis of the capacity of CD34⁺CD41⁺CD9⁻ and CD34⁺CD41⁺CD9⁺ to produce proplatelets and platelets*. i. Culture protocol of CD34⁺ cells and flow sorting. CD34⁺ cells cultured for 10 days in the

presence of SR1 were sorted according to their phenotype (CD34$^+$CD41$^{low}$, CD34$^+$CD41$^+$CD9$^-$ and CD34$^+$CD41$^+$CD9$^+$) using a FACS Aria II flow cytometer and then cultured for 6 days in a medium containing TPO with SR1 (left panel). ii. Representative differential interference contrast (DIC) microscopy photographs of a culture well at day 10+6 following cell sorting. Scale bar, 20 μm (right panel). iii. Number of platelets produced per well. CD34$^+$CD41$^+$CD9$^-$ cells able to produce platelets are called population of interest (POI). The cell suspension was subjected to multiple pipetting on day 10+6 of culture, and platelets were detected and counted by flow cytometry ($1.39 \pm 0.24 \times 10^6$ plts/well for CD34$^+$CD41$^{low}$ vs. $1.85 \pm 0.29 \times 10^6$ plts/well for POI vs. $1.6 \pm 0.60 \times 10^5$ plts/well for CD34$^+$CD41$^+$CD9$^+$) (mean $\pm$ SEM of 3 experiments; ** $p < 0.01$, ns $p > 0.05$, 1 way ANOVA and a Dunnett post test). (**C**) *Analysis of IKAROS expression.* i. Flow cytometric analysis of IKAROS (IKZF1) expression in POI compared to CD34$^-$CD41$^+$, mature megakaryocyte (MMK). ii. *Analysis of mean fluorescence intensity of IKZF1 in POI* vs. *MMK*. IKZF1 expression is increased by 3 in POI relative to MMK ($29.3 \pm 1.2$ IKZF1 MFI in MMK vs. $134.3 \pm 5.9$ IKZF1 MFI in POI) (mean $\pm$ SEM, n = 3, *** $p < 0.001$, Student $t$ test).

     We then analysed IKAROS expression in the POI by flow cytometry and observed a 4.5-fold higher expression compared to MMK ($134.3 \pm 5.9$ vs. $29.3 \pm 1.2$ MFI, respectively) (Figure 1Ci, ii). This pattern of IKAROS expression suggested its functional involvement in POI specification and expansion upon SR1 treatment. This was evaluated by combining gene silencing and pharmacological approaches. IKZF1 was knocked down using a specific shRNA lentivirally transduced in CD34$^+$ cells. A 75% decrease in IKZF1 expression was reached at day 10 in shIKZF1-transduced cells compared to cells transduced with a scramble shRNA ($17.4 \pm 2.2\%$ vs. $68.5 \pm 20.2\%$ expression, respectively) (Figure 2A), which resulted in a 3-fold decrease in POI generation ($12.5 \pm 0.7\%$ vs. $39.1 \pm 2.1\%$, respectively) (Figure 2B). In the second approach CD34$^+$ cells were treated with Lenalidomide (LenaL), a chemical compound inducing rapid and effective degradation of IKAROS (Figure S2). Exposure to increasing concentrations of LenaL in the presence of SR1 led to a gradual decrease in POI expansion, reaching a factor of 2 at 10 μM compared to control DMSO ($16.1 \pm 3.9\%$ vs. $32.9 \pm 2.5\%$, respectively) (Figure 2Ci). Together, these results provided evidence for a functional implication of IKAROS in the expansion of the POI and ensuing increased platelet production in response to SR1 ($9.44 \pm 2.1 \times 10^5$/well in SR1 vs. $4.1 \pm 1.5 \times 10^5$/well with LenaL 1 μM) (Figure 2Cii). To explore a possible molecular link between IKAROS and AHR in POI expansion we then performed a proximity ligation assay (PLA) [12], a fluorescence-based technique allowing the detection of protein-protein interaction based on physical proximity, using specific antibodies against AHR and IKAROS [12]. As shown in Figure 2Di, numerous positive signals were detected in the POI in response to SR1 whereas scarce positive signals were observed in control condition with DMSO ($7.8 \pm 1.0$ dots per POI with SR1 compared to $1.9 \pm 0.2$ dots in the control DMSO) (Figure 2Dii), indicating induction of nuclear AHR:IKAROS interaction in response to SR1.

     In a previous work, we demonstrated that co-culture of MKs with MSC also promoted the appearance of a CD34$^+$CD41$^{low}$ population through an AHR-dependent pathway, as shown by repression of *CYP1B1* [6]. Thus, we investigated whether this effect was, as for SR1, also dependent on AHR:IKAROS interaction. Beforehand, we established that CD34$^+$CD41$^+$CD9$^-$ and CD34$^+$CD41$^+$CD9$^+$ populations were also delineated under MSC co-culture conditions in proportions similar to those observed in SR1 condition, ($52.8 \pm 6.1\%$ of CD34$^+$CD41$^+$CD9$^-$ cells vs. $44.7 \pm 7.0\%$ of CD34$^+$CD41$^+$CD9$^+$ cells) (Figure 3A). Next, we observed that IKAROS was clearly better expressed in the POI ($78.2 \pm 4.6\%$ cells) than in MMKs ($45.9 \pm 7.1\%$ cells) (Figure 3). Additionally, IKZF1 shRNA knock down resulted in a markedly reduced POI amplification ($8.4 \pm 0.6\%$ in shIKZF1-derived POI vs. $36.2 \pm 2.5\%$ in shControl -derived POI) (Figure 3Ci–iii). Finally, as observed upon SR1 treatment, positive signals were observed using PLA indicating an AHR:IKAROS interaction in MSC-derived POI ($1.9 \pm 0.2$ dots/nucleus in control vs. $3.7 \pm 0.4$ dots/nucleus in MSC condition n = 81) (Figure 3D). Altogether, these results support the hypothesis that MSC, similarly to SR1, promotes POI amplification by acting on an AHR pathway that also involves a AHR:IKAROS interaction, thus distinct from the canonical pathway.

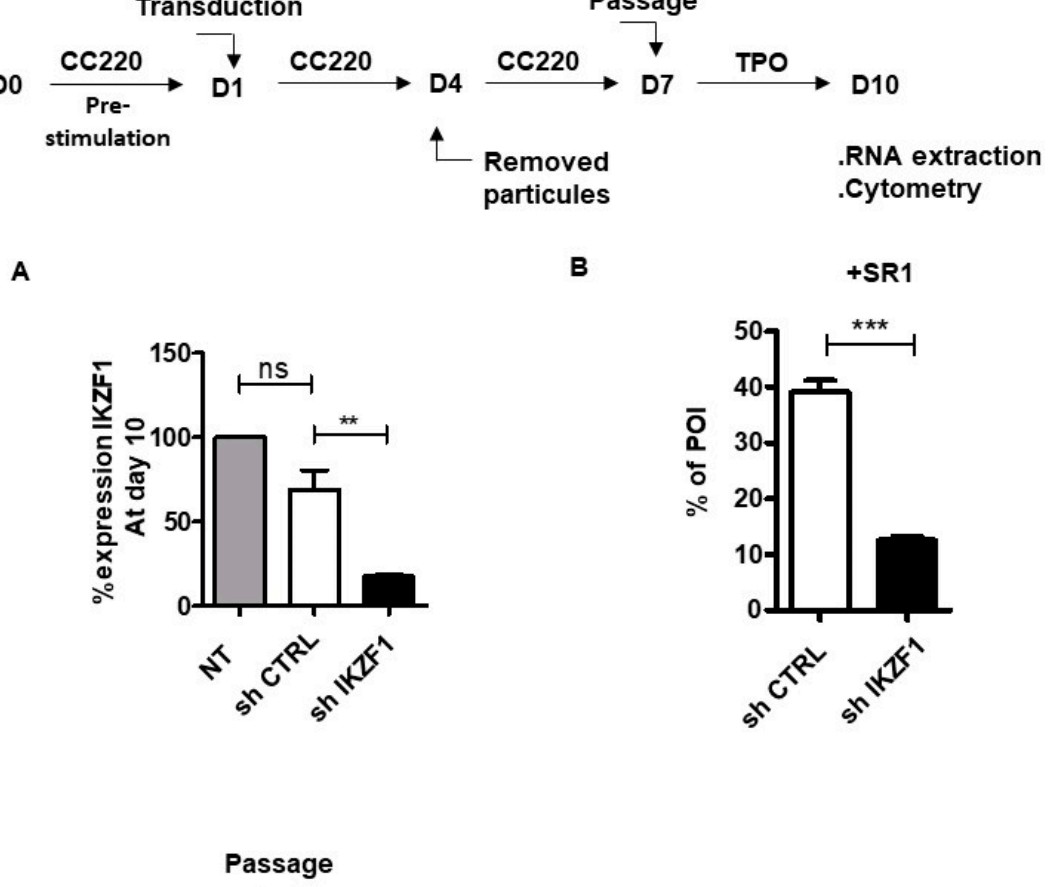

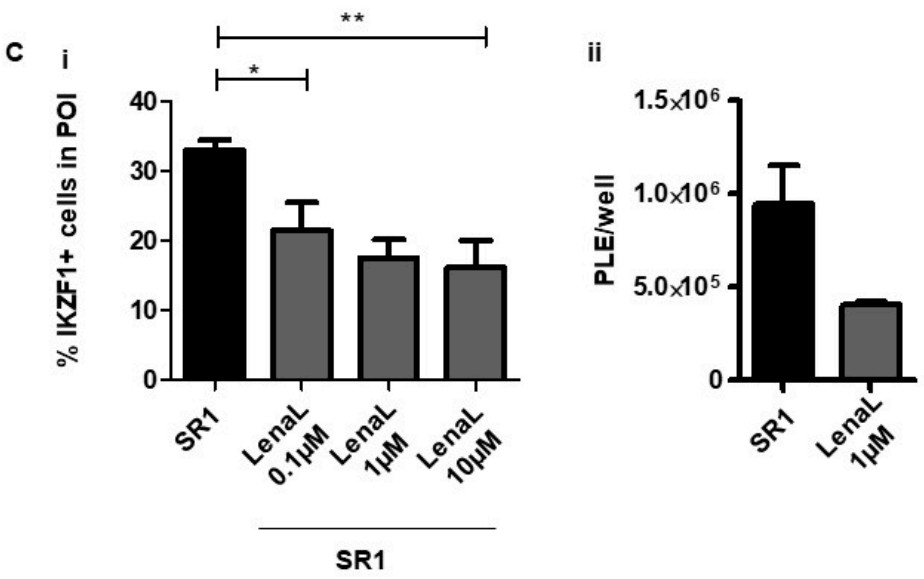

**Figure 2.** *Cont.*

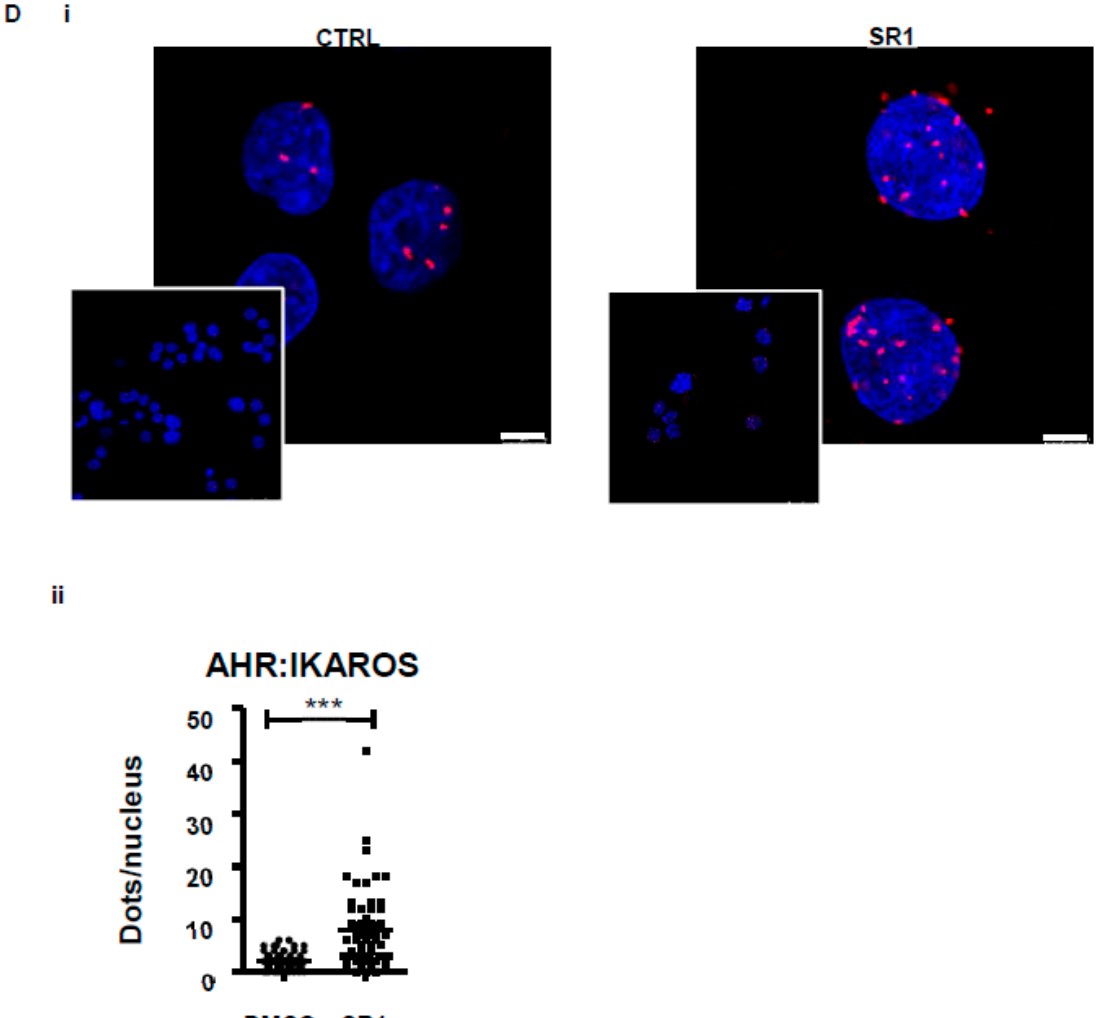

**Figure 2.** IKAROS: an actor of POI expansion. (**A**) *Validation of shIKZF1.* CD34+ cells were prestimulated with CC220 cytokine cocktail for 24 h and transduced with vectors expressing shControl GFP or IKZF1-specific shRNA and differentiated into MK for 6 days in a serum free medium containing CC220 cytokine cocktail and SR1, at day 7 cells were washed and then cultured for 3 supplemental days with SR1 and TPO. At day 10, total RNA derived from sorted transduced cells expressing GFP was subjected to qRT-PCR. The bar graph represents the percentage of IKZF1 expression in cells at day 10 ($68.7 \pm 11.9\%$ with sh scramble vs. $17.4 \pm 1.4\%$ with shIKZF1, the level of IKZF1 in NT cells has been arbitrary set at 100) (mean $\pm$ SEM, n = 3, ** $p < 0.01$, ns > 0.5, 1 way ANOVA and Bonferroni post test) (**B**) *shIKZF1-derived POI expansion.* Bar graph representing the % of POI-derived shIKZF1 or sh scramble transduced cells at day 10 ($39.1 \pm 2.1\%$ POI with sh scramble vs. $12.5 \pm 0.7\%$ POI with shIKZF1) (mean $\pm$ SEM, n = 3, *** $p < 0.001$, Student *t* test) (**C**) *Effect of Lenalidomide (LenaL) on POI expansion.* CD34+ cells were cultured for 7 days in serum free medium in presence of SR1 and CC220 cytokine cocktail. At day 7, cells were washed and cultured for 3 supplemental days with TPO and SR1. LenaL was added at increasing concentrations (0.1 μM, 1 μM and 10 μM) at day 0 and again at day 7. (i) Bar graphs representing percentage of IKAROS-positive cells in POI ($21.4 \pm 4.1$ with LenaL 0.1 μM vs. $17.5 \pm 2.6$ with LenaL 1 μM $16.1 \pm 3.9$ % with LenaL 10 μM) (mean $\pm$ SEM of three experiments; * $p < 0.05$, ** $p < 0.01$, 1 way ANOVA and a Dunnett post test). (ii) Bar graphs representing the number of platelets released per well ($9.4 \pm 2.1 \times 10^5$ vs. $4.1 \pm 1.5 \times 10^5$, ns $p > 0.05$, n = 3 Student *t* test) (**D**) (i) Representative images of the interaction between IKAROS and AHR in cells at day 10 examined by proximity ligation assay (PLA), where the interaction is visualized by red dots, scale bar: 5 μm. (ii) Dot plots representing the number of positive dots/nucleus in POI obtained in response to SR1 or in DMSO conditions. Positive signals were counted on eight images in 50 cells ($7.8 \pm 1.0$ dots per POI with SR1 compared to $1.9 \pm 0.2$ dots in the control DMSO, n = 2, *** $p < 0.001$).

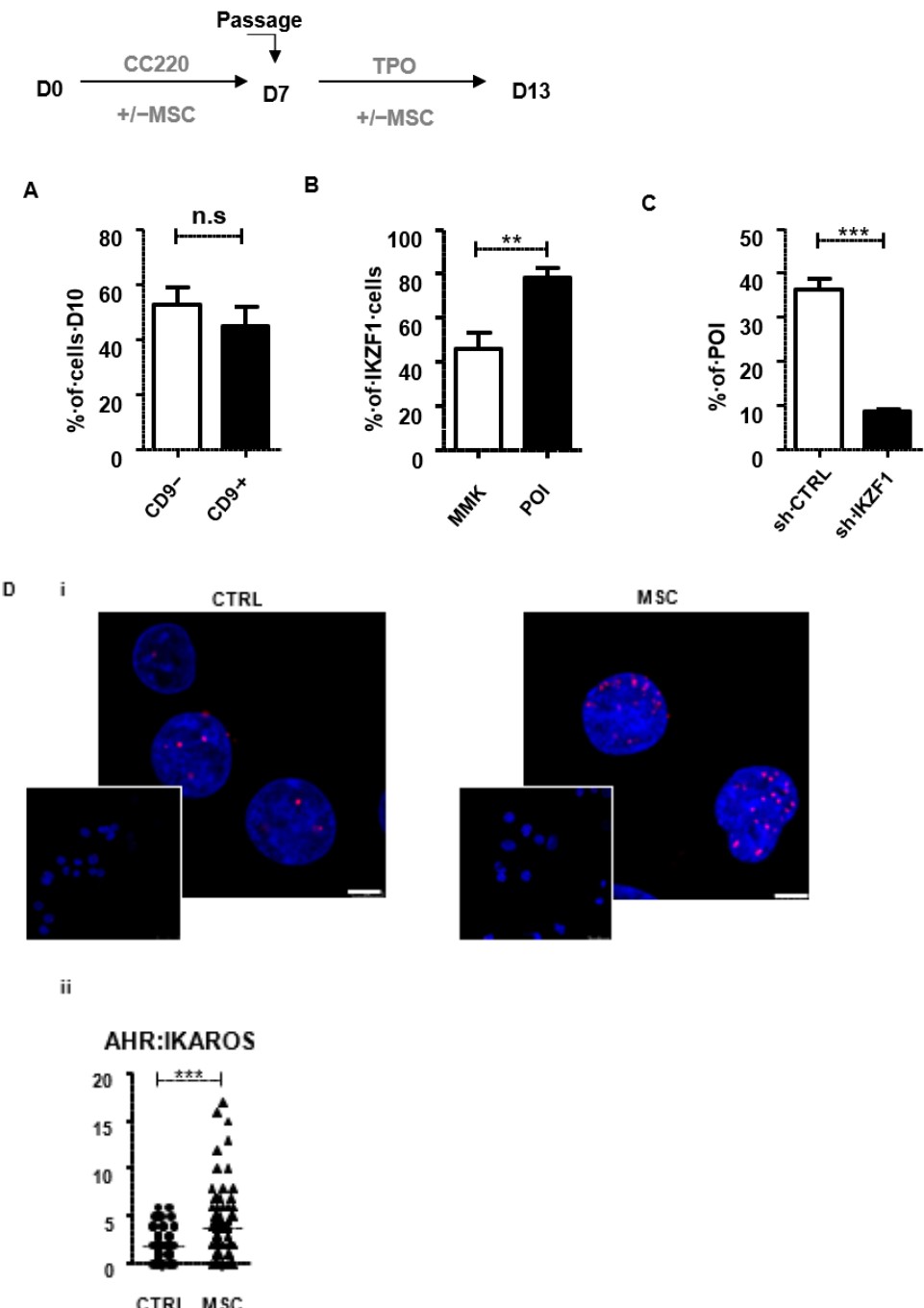

**Figure 3.** The AHR:IKAROS interaction is involved in MSC-triggered POI amplification. Peripheral blood CD34$^+$ cells were cultured in the absence or presence of monolayer of MSC in serum free medium containing a CC220 cytokine cocktail from day 0 to 7 and with TPO from day 7 to 10. (**A**) Bar graphs representing the percentage of CD9$^+$ and CD9$^-$ subpopulations within the CD34$^+$CD41$^{low}$ population obtained at day 10 and analysed by flow cytometry (52.8 $\pm$ 6.1% of CD34$^+$CD41$^+$CD9$^-$ cells vs. 44.7 $\pm$ 7.0% of CD34$^+$CD41$^+$CD9$^+$ cells) (mean $\pm$ SEM, n = 3, ns $p$ > 0.05, Student $t$ test) (**B**) Bar graphs representing the percentage of IKZF1 expression in MSC-derived POI or MSC-derived MMK (78.2 $\pm$ 4.6 in POI vs. 45.9 $\pm$ 7 in MMK) (mean $\pm$ SEM, n = 3, ** $p$ < 0.01, Student $t$ test). (**C**) Bar graphs representing the percentage of MSC-derived POI obtained following CD34$^+$ cells transduction with shIKZF1 or sh scramble (8.4 $\pm$ 0.6 of shIKZF1-derived POI vs. 36.2 $\pm$ 2.5 sh scramble -derived POI) (mean $\pm$ SEM, n = 3, *** $p$ < 0.001, Student $t$ test) (**D**) (i) Representative images of interaction between AHR and IKAROS analyzed by PLA in MSC-derived POI and in control POI without MSC, scale bar: 5 μm. (ii) Dot plots representing the number of positive signals/nucleus in POI obtained in response to CTRL or in MSC conditions. Positive signals were counted on 10 images 50 cells (1.9 $\pm$ 0.2 signals in the control vs. 3.7 $\pm$ 0.4 signals per POI with MSC, *** $p$ < 0.001).

Our previous work highlighted the existence and expansion of a megakaryocyte progenitor with high platelet production capacity in response to an AHR antagonist [6]. Repression of the target gene, *CYP1B1* in response to SR1 and also after co-culture of CD34+ cells with MSC initially raised the possibility that this could occur via the canonical pathway. The present study revealed an alternative mechanism via the interaction between AHR and IKAROS, supported by the following observations of (i) increased IKAROS expression in the POI, (ii) prevention of POI generation following down-regulation or degradation of IKAROS, (iii) demonstration of a physical interaction between AHR and IKAROS, and (iv) negative regulation of megakaryocytic TFs.

In addition to highlighting a new mechanism that regulates megakaryopoiesis, our work nicely connects previous reports on the role of IKAROS in MK specification and maturation [8,9] and AHR:IKAROS interaction in ILC3 cells [10]. This work opens the possibility of genetic or pharmacological intervention on this alternative pathway to improve our capacity to produce cultured platelets at more acceptable costs.

## 3. Material and Methods

### 3.1. Extraction of CD34+ Cells

CD34+-enriched cells were obtained from leukofilters (TACSI, Terumo, Tokyo, Japan by magnetic-activated cell sorting (CD34 MicroBead Kit UltraPure, Miltenyi Biotec, Bergisch Gladbach, Germany) as described [6] The mean percentage of viable CD34+ cells was $97.9 \pm 0.3\%$, n = 13.

### 3.2. Megakaryocyte Differentiation in Culture

In the first phase, CD34+ cells were seeded in 24-well plates at a density of $4 \times 10^4$/mL in StemSpan Serum-Free Expansion Medium (SFEM) supplemented with 20 µg/mL human LDL, a cocktail of cytokines containing SCF, TPO, IL-6 and IL-9 (StemSpan™ Megakaryocyte Expansion Supplement) and 1 µM SR1 (all from Stemcell Technologies, Vancouver, Canada). On day 7, the cells were harvested, washed and seeded at a density of $5 \times 10^5$/mL in StemSpan SFEM containing 1 µM SR1, 50 ng/mL TPO and 20 µg/mL human LDL and cultured for an additional 6 days.

For CD34+ and MSC co-culture, cells were seeded on a confluent layer of MSCs at day 0 and day 7 [13].

The inducer of IKAROS degradation lenalidomide (gift from Chan S. lab) was added in the culture at concentrations ranging from 0.1 µM to 10 µMat at days 0 and 7.

### 3.3. Lentiviral Short Hairpin Knockdown
Vectors Used

shRNA constructs targeting the human IKZF1 and shRNA scramble control cloned intro psi-LVRU6GP were obtained from GeneCopoeia (Rockville, MD, USA). The plasmids pCG-HΔ24 and pCG-FΔ30 encoding for glycoprotein of measles virus envelope were kindly provided by Verhoeyen E [14] and p8.91, an encapsidation plasmid lacking all accessory HIV-1 proteins (Vif, Vpr, Vpu and Nef) by A. Dubart-Kupperschmitt.

### 3.4. Lentivirus Production and CD34+ Cells Trasnduction

Lentiviruses were produced by transient transfection of HEK293T cells in DMEM (Invitrogen, Carlsbad, CA, USA).

*Plasmids*: pCG-HΔ24 (5 µg), pCG-FΔ30 (5 µg), p.8.91 (16 µg), psi-LVRU6GP (IKZF1 or shControl) (16 µg) were transfected with Lipofectamine 2000 (100 µL) in Optimem medium (Invitrogen) into HEK293T cells. After 18 h of transfection, the medium was replaced by Opti-MEM (Invitrogen). Two days later, viral supernatants were harvested and filtered. They were concentrated by ultracentrifugation and the resulting pellet was suspended in phosphate buffered saline (PBS) and frozen at −80 °C in 25-µL aliquots until use.

*Infectious titers* (in transduction units [TU]/mL) were determined by flow cytometry analysis of target cells using serial dilutions of the supernatants added to HEK293T.

*CD34⁺ cells transduction*. CD34⁺ cells were grown for 24 h in Stemspan medium with StemSpan™ Megakaryocyte Expansion Supplement (Stemcell Technologies). Then, cells were thoroughly washed and seeded in wells coated with retronectin (Takara Bio, Kusatsu, Japan). Transduction was subsequently performed by adding lentiviral particles at 5 MOI for 72 h. At days 7 and 10, following infection, cells were analyzed by flow cytometry.

### 3.5. Cell Sorting and Flow Cytometry Analysis

*Cell sorting*. Cell sorting was performed using an Aria III flow cytometer as previously described by our group [6]. Antibodies are listed in Table 1. The sorted CD34⁺41$^{low}$CD9⁻ and CD34⁺41$^{low}$CD9⁺ populations were seeded in Stemspan medium containing TPO 50 ng/mL, LDL 20 µg/mL and SR1 1 µM for 7 day.

**Table 1.** Antibodies used in different experiments.

| Target | Clone | Labelling | Host | Applications | Supplier |
|--------|-------|-----------|------|--------------|----------|
| CD41/61 | ALMA17 | Alexa Fluor 488 or 647 | Mouse | Flow cytometer | EFS-Grand Est |
| CD34 | 4H11 | PECY7 | Mouse | Flow cytometer | ThermoFisher Scientific |
| CD9 | HI9a | PE | Mouse | Flow cytometer | Biolegend |
| IKZF1 | 16B5C71 | Alexa Fluor 647 | Mouse | Flow cytometer | Biolegend |
| IgG2a | MOPC-173 | Alexa Fluor 647 | Mouse | Flow cytometer | Biolegend |
| IKZF1 | | Unconjugated | Rabbit | Flow cytometer/PLA | Gift Chan S. lab |
| AHR | 3B12 | Unconjugated | Mouse | PLA | Sigma-Aldrich |

*Flow cytometry analysis*. Flow cytometry analysis was performed using Fortessa X20 (Becton Dickinson, Franklin Lakes, NJ, USA), as previously described by [6]. Antibodies are listed in Table 1.

For intracellular staining, cells were first labeled with surface antibodies, fixed, permeabilized using the Foxp3/Transcription Factor Staining Buffer Set (Invitrogen) according to the manufacturer's instructions and then labeled with an anti-IKAROS antibody or mouse IgG2a κ-647 (BioLegend, San Diego, CA, USA) (Table 1).

*Platelet generation*. Culture-derived platelets were released following successive pipetting after the addition of 0.5 µM PGI2 and 0.02 U/mL apyrase in the culture medium and analyzed as previously described [6].

All samples were analyzed using a Fortessa X20 flow cytometer (Becton Dickinson).

### 3.6. Quantitative RT-PCR

RNA extraction and cDNA synthesis were performed using RNeasy Mini Kit (Qiagen, Hilden, Germany) and RT2 First Strand Kit (Qiagen), respectively, according to manufacturer's instructions. qPCR was performed using the RT² SYBR Green qPCR Mastermix (Qiagen) on a CFX96 Touch Real-Time PCR Detection System (Biorad, Hercules, CA, USA). Relative transcript levels were calculated by the method of ΔCt with TBP like reference gene. Primers set for *TBP*, *IZKF1*, *ZFMP2*, *CD9*, *PF4*, *GP1bA*, *NF-E2*, *GATA1* were purchased from Genecopoeia.

### 3.7. DuoLink Proximity Ligation Assay (PLA) Analysis

CD34⁺-derived megakaryocytes were harvested and cytospun onto poly-L-lysine coated slides. Immobilized cells were then fixed with PFA 4–0.01% glutaraldehyde and permeabilized with 0.05%Triton X100. PLA was subsequently performed according to manufacturer's instructions using primary antibodies against IKAROS and AHR described in Table 1. The interacting tandems AHR/IKAROS were visualized by fluorescence on a SP8 confocal microscope (Leica, Wetzlar, Germany).

### 3.8. Statistical Analyses

Results were expressed as the mean ± SEM and statistical comparisons were performed using an unpaired, two-tailed Student's *t*-test or one-way ANOVA followed by the



Bonferroni post-hoc test (Prism, Graph-Pad Software Inc., San Diego, CA, USA, version 5). *p* values of less than 0.05 were considered to be statistically significant.

**Supplementary Materials:** The following are available online at https://www.mdpi.com/2571-841 X/4/1/7/s1, Figure S1: Analysis of TF expression in POI, Figure S2: Analysis of cells positive for Ikaros in presence of Lenalidomide.

**Author Contributions:** C.S., H.d.l.S. and F.L. conceived and designed the experiments; L.M. and V.D.S. performed the experiments; C.S., L.M. and V.D.S. analyzed the data; C.S., L.M., V.D.S., S.C., P.K., H.d.l.S., C.G. and F.L. discussed the techniques and results; S.C. and P.K. provided Lenalidomide and anti-Ikaros antibody. C.S. supervised the work; and C.S., H.d.l.S. and F.L. wrote the paper. All authors have read and agreed to the published version of the manuscript.

**Funding:** This research was funded by ANR-17-CE14-0001-1.

**Institutional Review Board Statement:** Human studies were performed according to Helsinki declaration. Healthy blood donors were recruited by Etablissement Français du Sang-Grand Est, where the research was performed. These blood gifts, dedicated to blood transfusion, follow a processing step that eliminate the leukocytes. The cells used for this study are recovered from leukoreduction filters that are normally destroyed. The volunteers gave written informed consent stating that part of their blood gift may be used for research.

**Informed Consent Statement:** All procedures were registered and approved by the French Ministry of Higher Education and Research and registered under the number AC_2015_2371.The donors gave their approval in the CODHECO number AC_2008_562 consent form, in order for the samples to be used for research purposes.

**Data Availability Statement:** No Applicable.

**Conflicts of Interest:** The authors declare no competing financial interests.

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
