# Peer review of "AHR:IKAROS Interaction Promotes Platelet Biogenesis in Response to SR1"

_reports, doi:10.3390/reports4010007_

Round 1

Reviewer 1 Report

In the short report “AHR:IKAROS interaction promotes platelet biogenesis in response to SR1”, the authors showed that the emergence of the thrombocytogenic precursors and the enhancement of platelets production triggered by SR1 involved Ikaros (IKZF1).

It contributes to an innovative acknowledgement about AhR/Ikaros-dependent pathway which prompt the expansion of the thrombocytogenic precursors. The contents fit into the field and the aim of this journal.

I suggest that the short report could be considered suitable for publication in the current form.

Author Response

Dear reviewer, thank you for your acceptance

best regards

Catherine Strassel

Reviewer 2 Report

In the short report "AHR:IKAROS interaction promotes platelet biogenesis in response to SR1" the authors provide a AHR/Ikaros-dependent regulation of SR1-induced platelet biogenesis. The study and results are of interest and have impact in the field of in vitro megakaryocyte differentiation. Although, the experiments are well performed, the text and especially the figures including the figure legends have to be carefully checked and improved. The manuscript was not prepared with meticulous care and carefully proof read. Moreover, some additional experiments should be performed to strengthen the manuscript.  

Major points:

  1. The authors do not adequately discuss their results in the context of recent findings and studies. For example, is there anything known about downstream effector kinases of IKAROS playing a role in megakaryopoiesis? Please adjust and improve the discussion of the results of the manuscript.
  2. Are there different expression levels of the c-mpl (CD110) receptor in the different subpopulations of megakaryocytes? TPO is the major compound inducing and regulating the platelet biogenesis. So it is crucial to investigate at least the expression of the TPO receptor levels in CD9- and CD9+ cells, since a difference could explain the different maturation of the cells.
  3. Lenalidomide is known to be able to induce apoptosis. Does it induce apoptosis in megakaryocytes? Therefore, the authors have to investigate the apoptotic state of megakaryocytes. Annexin-V or DNA fragmentation measurements could be performed.
  4. It is difficult to interpret the images provided in Figure 1 Bii. So please provide better quality and bigger images. Moreover, a statistical analysis of the pro-platelets formation has to be added to clarify the conclusions.
  5. The authors should clearly state if they used biological or technical repeats throughout the manuscript, since only maximum 3 repeats were performed which could undermine the conclusions.
  6. In Figure S1 the label of the y axis is missing and has to be added.

Text/Figure organization and spelling:

  1. In the abstract the terms and the abbreviations are always provided. So please provide the term for the SR1 abbreviation (line 15).
  2. The subheading of the main text body seem not appropriate for a short report. Is the subheading "Introduction" correct since it also includes the results and discussion? Please check and possibly change.
  3. The sentence in line 40-42 is a bit confusing and contains not known characters and a redundant bracket. So please correct and improve.
  4. Please correct "zinc-finger" in line 54.
  5. Line 56: please remove the redundant dots.
  6. The term POI is a little bit confusing. Why not term it CD34+CD9-CD41?
  7. Figure legend of Figure 1 does not contain B). Most likely B) should be added in line 108.
  8. Why the figure legend (line 112-114) is all of a sudden in italic? Is there any reason?
  9. Why the figure legend of figure 2 and 3 has a different font? Please adjust.
  10. Please add a space in line 130 before "population".
  11. The Figure 3 and the corresponding legend does not fit. While in the legend it is called E, in the figure it is labeled Dii. Please check and improve.
  12. The heading of the legend of Figure S2 is missing.

Author Response

Please find my reply to your comments

Major points:

  1. The authors do not adequately discuss their results in the context of recent findings and studies. For example, is there anything known about downstream effector kinases of IKAROS playing a role in megakaryopoiesis? Please adjust and improve the discussion of the results of the manuscript.

As requested by the reviewer, elements have been added to the text (P2, l 45-18), and highlighted in yellow.

  1. Are there different expression levels of the c-mpl (CD110) receptor in the different subpopulations of megakaryocytes? TPO is the major compound inducing and regulating the platelet biogenesis. So it is crucial to investigate at least the expression of the TPO receptor levels in CD9- and CD9+ cells, since a difference could explain the different maturation of the cells.

As mentioned by the reviewer, TPO is well known as a crucial regulator of platelet biogenesis and expression of its receptor, mpl, could influence the maturation of CD9- and CD9+ population. Based on a transcriptome that we have performed on the CD9+ and CD9- populations (manuscript in progress), we did not observe difference in the level of expression of mpl (see figure 1 for referee). These observations indicate that the difference in the ability of cells to produce platelets is not directly linked to their level of mpl expression.

  1. Lenalidomide is known to be able to induce apoptosis. Does it induce apoptosis in megakaryocytes? Therefore, the authors have to investigate the apoptotic state of megakaryocytes. Annexin-V or DNA fragmentation measurements could be performed.

As requested by the Reviewer, the apoptotic state of MK has been analyzed by looking at the appearance of Annexin V and caspase 3 in the presence of increasing concentrations of lenalidomide (0.1-10 µM). As shown in Figure 2 for referee, flow cytometry analysis shows no difference in in annexin V staining and caspase 3 activity in the MK population.

  1. It is difficult to interpret the images provided in Figure 1 Bii. So please provide better quality and bigger images. Moreover, a statistical analysis of the pro-platelets formation has to be added to clarify the conclusions.

As requested, better quality images are now provided in Figure 1Bii. Although the proportion of proplatelets is very often provided in the literature to illustrate the efficiency of thrombopoiesis in culture conditions, it is our experience that its determination is somewhat imprecise and operator-dependent. In our view, the number of released platelets is a more precise and robust parameter to evaluate the efficiency of thrombopoiesis, reflecting the MK's ability to release morphologically and functionally bona fide platelets, as recently shown in our recent publication (Do Sacramento et al, 2020). 

  1. The authors should clearly state if they used biological or technical repeats throughout the manuscript, since only maximum 3 repeats were performed which could undermine the conclusions.

Throughout the manuscript we used biological repeats.

  1. In Figure S1 the label of the y axis is missing and has to be added.

As mentioned by the reviewer, the label of the y axis has been added.

Figure organization and spelling:

  1. In the abstract the terms and the abbreviations are always provided. So please provide the term for the SR1 abbreviation (line 15).

The term related to the abbreviation SR1, Stem-Regenin 1, has been provided, line 18 and highlighted in yellow.

  1. The subheading of the main text body seem not appropriate for a short report. Is the subheading "Introduction" correct since it also includes the results and discussion? Please check and possibly change.

A section including the subheading: Result and Discussion has been added, line 50 and highlighted in yellow.

  1. The sentence in line 40-42 is a bit confusing and contains not known characters and a redundant bracket. So please correct and improve.

The section has been improved.

  1. Please correct "zinc-finger" in line 54.

The correction has been done.

  1. Line 56: please remove the redundant dots.

The redundant dots have been removed

  1. The term POI is a little bit confusing. Why not term it CD34+CD9-CD41?

We have chosen the term POI "population of interest" because this population has a unique ability to extend proplatelets. It is also the population of interest in this paper as understanding how this population emerges and expands is paramount to improving the manufacture of therapeutic cultured platelets. 

  1. Figure legend of Figure 1 does not contain B). Most likely B) should be added in line 108.

The legend has been corrected.

  1. Why the figure legend (line 112-114) is all of a sudden in italic? Is there any reason?

As requested, the typography has been respected.

  1. Why the figure legend of figure 2 and 3 has a different font? Please adjust.

As requested, the legend front has been adjusted.

  1. Please add a space in line 130 before "population".

It has been corrected

  1. The Figure 3 and the corresponding legend does not fit. While in the legend it is called E, in the figure it is labeled Dii. Please check and improve.

It has been corrected.

  1. The heading of the legend of Figure S2 is missing.

The Heading has been added

Reviewer 3 Report

This is an interesting research study focusing on the production
of cultured platelets with great clinical impotance. The study is well-planned and proper methds are used (as previously shown by the same workgroup). 

Author Response

Dear reviewer

thank you for your acceptance

best regards

Catherine strassel